# Cleavage of MALAT1 RNA by 14-nt sgRNA-guided tRNase Z^L

**Masayuki Takahashi, Masayuki Nashimoto**◉*

RNA Therapeutics Laboratory, Faculty of Medical Technology, Niigata University of Pharmacy and Applied Life Sciences, Niigata, Japan

* mnashimoto@nupals.ac.jp

## Abstract

We have been developing a gene suppression technology, tRNase Z^L-utilizing efficacious (TRUE) gene silencing, in which artificially designed small guide RNA (sgRNA) guides tRNase Z^L to cleave cellular target RNA. In this study, we examined 14-nt linear-type sgRNAs, which are fully 2′-O-methylated and have full phosphorothioate linkages, for their ability to suppress a level of a nuclear-localized long non-coding RNA, Metastasis Associated Lung Adenocarcinoma Transcript 1 (MALAT1). The MALAT1 RNA is implied to be involved in stress responses and diseases including cancers. Specifically, we designed six 14-nt linear-type sgRNAs, sgRM1−sgRM6 that target the human MALAT1 RNA. sgRM1, sgRM2 and sgRM6 suppressed the MALAT1 RNA level, while the other sgRNAs showed little effect. In order to demonstrate that the suppression effect of sgRM1, sgRM2 and sgRM6 on the MALAT1 RNA level is caused by TRUE gene silencing, we performed *in vitro* tRNase Z^L cleavage assay, microscopic analysis for nuclear existence of sgRNA, and tRNase Z^L knockdown experiment. For the *in vitro* tRNase Z^L cleavage assay, three 30-nt MALAT1 RNA fragments, TM1, TM2 and TM6 were prepared, which were RNA targets for sgRM1, sgRM2 and sgRM6, respectively. All of the sgRNAs guided recombinant tRNase Z^L *in vitro* to cleave their own targets, although the cleavage efficiency changed depending on target/sgRNA pairs. By fluorescence microscopy, a 14-nt 5′-Alexa568-labeled sgRNA released from liposome was observed to be distributed ubiquitously in A549 cells with higher density in the nucleus, where both the target MALAT1 RNA and tRNase Z^L exist. Knockdown of tRNase Z^L by siRNA attenuated the suppression effect of sgRM1, sgRM2 and sgRM6 on the MALAT1 RNA level. We also demonstrated that the effective sgRNAs sgRM1, sgRM2 and sgRM6 reduce A549 cell viability.

**Data availability statement:** All relevant data are within the manuscript and its Supporting Information files.

**Funding:** This work was supported by grant number 156 (to MN) from Takahashi Industrial and Economic Research Foundation. The funders had no role in study design, data collection and analysis, decision to publish, or preparation of the manuscript.

**Competing interests:** I have read the journal's policy and the authors of this manuscript have the following competing interests: The author MN is an advisor of Veritas In Silico Inc., and owns stock of the company.

## Introduction

The discovery in 1991 of a sequence-specific endoribonuclease activity that needs a cellular tRNA fragment led us to attempt to develop a gene suppression technology, tRNase $Z^L$-utilizing efficacious (TRUE) gene silencing [1–3]. TRUE gene silencing is based on the property of tRNase $Z^L$ that it can recognize and cleave a pre-tRNA-like or micro-pre-tRNA-like complex (S1 Fig.) [3]. Such a complex can be formed between a target RNA and an artificially designed small guide RNA (sgRNA), which is categorized into four groups, 5′-half-tRNA-type, heptamer-type, hook-type, and ~14-nt linear-type. The way to design hook-type and linear-type sgRNAs is straightforward, whereas we need to find a tRNA T-arm-like structure in a target RNA to design 5′-half-tRNA-type and heptamer-type sgRNAs. We have intensively studied RNA structures that can be recognized by tRNase $Z^L$, and shown that any target RNA can be cleaved at any desired site by tRNase $Z^L$ with the aid of appropriately designed sgRNA at least in a test tube [3]. Although appearing to have a clinical potential, this technology has been progressing slowly and is still in its infancy [4–6].

There are two well-established technologies to suppress gene expression [7]. One uses conventional antisense oligonucleotides (ASOs) that can guide RNase H1 to cleave target RNA and the other uses small interfering RNAs (siRNAs) that can guide Argonaute 2. And some of the conventional ASOs and the siRNAs have been approved as therapeutic drugs. Another new technology for gene silencing based on prokaryotic immune systems is emerging and appears to be promising [8,9]. This technology utilizes clustered regularly interspaced short palindromic repeats RNAs that can guide Cas13 or Csm to cleave target RNA. Under such circumstances, one may think that clinical applications of TRUE gene silencing would not be worth pursuing. However, we believe that we would have a chance to find a niche of this technology, since there would be a large number of potential target RNAs that cause various diseases and a subset of them may not be able to be eliminated easily by the above established and emerging technologies.

We have been attempting to apply TRUE gene silencing to multiple myeloma or leukemia to find a cure by searching for effective sgRNAs [10–15]. In this attempt, among the four types, heptamer-type sgRNAs were chosen, which were 5′-/3′-phosphorylated and fully 2′-O-methylated, considering cost performance, nuclease resistance, and the efficiency in guiding tRNase $Z^L$ [16]. We have recently, however, shown that longer oligonucleotides are intracellularly taken up more efficiently and that the oligonucleotides with phosphorothioate linkages are taken up by cells more efficiently than those without the linkages [17]. Taking account of these observations and the easiness of design, here, we examined 14-nt linear-type sgRNAs, which are 5′-/3′-phosphorylated and fully 2′-O-methylated and have full phosphorothioate linkages, for their ability to suppress a level of a long non-coding RNA (lncRNA), Metastasis Associated Lung Adenocarcinoma Transcript 1 (MALAT1).

The MALAT1 RNA, which is localized in the nucleus, is implied to be involved in stress responses and diseases including cancers [18]. And downregulation of the MALAT1 RNA level by conventional ASOs has been shown to suppress metastasis of human lung cancer cells or growth of human myeloma cells in a mouse xenograft

model [19,20]. Since the conventional ASO technology is established and works well as above, in this study, we focused on examining how 14-nt linear-type sgRNAs affect the cellular level of nuclear-localized lncRNA and finding a way to improvement of TRUE gene silencing rather than comparing with the efficacy of the ASO technology.

## Materials and Methods

### Oligonucleotide preparation

The following oligoribonucleotides were chemically synthesized and subsequently purified by high-performance liquid chromatography by Nippon Bioservice (Saitama, Japan): six 5′-/3′-phosphorylated, fully 2′-O-methylated sgRNAs with full phosphorothioate modifications, sgRM1 (5′-UAGGAUUCUAGACA-3′), sgRM2 (5′-UGGUUAUGACUCAG-3′), sgRM3 (5′-AUUGCCUCUUCAUU-3′), sgRM4 (5′-CCUUCUGCCUUAGU-3′), sgRM5 (5′-CUUUUGCAUUUCCC-3′), sgRM6 (5′-AAUCCCCUAGGGAA-3′), sgNC1 (5′-CCCCCCCCCCCCCC-3′) and sgNC2 (5′-CUCUCUCUCUCUCU-3′), and a 5′-Alexa568-labeled, 3′-phosphorylated, fully 2′-O-methylated sgRNA with full phosphorothioate linkages (sgRTEST, 5′-UUGGUACCUUCUCA-3′).

The following primers used for quantitative reverse-transcription polymerase chain reaction (qRT-PCR) were obtained from Fasmac (Kanagawa, Japan): the primer pair pp0 of MALAT1 forward primer-0 (5′-GGATCCTAGACCAGCATGCC-3′) and MALAT1 reverse primer-0 (5′-AAAGGTTACCATAAGTAAGTTCCAGAAAA-3′), the primer pair pp1 of MALAT1 forward primer-1 (5′-GCAAAATATGTTTTCAGTTC-3′) and MALAT1 reverse primer-1 (5′-ACCTGTGGTGGTCTGATTAT-3′), the primer pair pp2 of MALAT1 forward primer-2 (5′-CTTTTGTTGATGAGGGAGGG-3′) and MALAT1 reverse primer-2 (5′-TTTCTCTGCATCTAGGCCAT-3′), the primer pair pp6 of MALAT1 forward primer-6 (5′-AAGTGCTTATTTTAA GAGC-3′) and MALAT1 reverse primer-6 (5′-TTAAAAAGGCTCGATGGAAA-3′), β-actin forward primer (5′-ACAATGTGG CCGAGGACTTT-3), and β-actin reverse primer (5′-TGTGTGGACTTGGGAGAGGA-3′).

The following primers used for 3′ rapid amplification of cDNA ends (3′-RACE) were obtained from Fasmac: primer-RT (5′-AAAAGCGGCCGCCTGGAATTCGCGGTTAAA-3′), FWD11 (5′-AAAACTCGAGTTTATTTTCTAATATAATGGGG GAG-3′), FWD12 (5′-AAAACTCGAGGATGGCAAGTTTGTGGGTTTTTT-3′), FWD13 (5′-AAAACTCGAGTAATGAAG TATTTCAGTTTTGTG-3′), FWD21 (5′-AAAACTCGAGACTGGGCTGACATTAACTACAATTA-3′), FWD22 (5′-AAAACTC GAGCTGATGGTAGCTTTTGTATTATC-3′), FWD23 (5′-AAAACTCGAGGTATGCTTCAAAAATTTTGTAAA-3′), FWD61 (5′-AAAACTCGAGGAAAGCTACCAATTTAAAGTTACGG-3′), FWD62 (5′-AAAACTCGAGGTTACACTACATTAATCCTG GAA-3′), and FWD63 (5′-AAAACTCGAGCTCTTCGTTATCAGAAGAGTTGC-3′). 3′-RACE-linker (5′-r(UUU)d(AACCGC GAATTCCAG)-3′) that was 5′-/3′-phosphorylated was obtained from Nippon Bioservice.

Three 5′-carboxyfluorescein (FAM) labeled MALAT1 RNA fragments, TM1 (5′-GGUCUGUCUAGAAUCCUAAAGG CAAAUGAC-3′), TM2 (5′-UCUUCUGAGUCAUAACCAGCCUGGCAGUAU-3′), and TM6 (5′-CCCCUUCCCUAGGG GAUUUCAGGAUUGAGA-3′) were chemically synthesized and subsequently purified by high-performance liquid chromatography by Ajinomoto Bio-Pharma (Tokyo, Japan).

siRNAs, sitRNase Z$^L$ (sense strand sequence: 5′- CGCUGUUGCGAACAUGUGAUU-3′) and siControl (sense strand sequence: 5′-CAGCACGACUUCUUCAAGUCC-3′) were obtained from Nippon Bioservice (Saitama, Japan).

### Calculation of melting temperature (Tm) and RNA secondary structure

Tm was calculated by the formula 4°C × (number of G's + C's in the sgRNA) + 2°C × (number of A's + U's in the sgRNA). Potential RNA secondary structure was calculated on the website http://www.unafold.org.

### Cell culture

Human A549 cells [21] and HEK293 cells [22] were cultured in DMEM (Wako, Osaka, Japan) supplemented with 10% fetal bovine serum (COSMO BIO, Tokyo, Japan) and at 37°C in a humidified incubator with 5% $CO_2$.

## Transfection

Transfection was performed using Lipofectamine 3000 (Thermo Fisher Scientific, Tokyo, Japan) following the manufacturer's protocol for reverse transfection. In tRNase $Z^L$ knockdown experiments, A549 cells were transfected with 20 nM siRNA, and after 24-hr culture, they were transfected with 200 nM sgRNA and further cultured for 96 hr.

## qRT-PCR

Total RNA was extracted from A549 cells using TRI Reagent (Molecular Research Center, Cincinnati, USA), and cDNA was synthesized using PrimeScript RT Reagent Kit (Perfect Real Time) (Takara, Shiga, Japan) according to the manufacturer's protocol. MALAT1 RNA, tRNase $Z^L$ mRNA and β-actin mRNA were quantified by PCR with THUNDERBIRD SYBR qPCR Mix (TOYOBO, Osaka, Japan) using a Thermal Cycler Dice Real Time System (Takara).

## Preparation of recombinant human tRNase $Z^L$

The histidine-tagged human tRNase $Z^L$ that lacks N-terminal 30 amino acids was overexpressed using the expression plasmid pQE-80L in *E. coli* strain Rosetta(DE3)pLysS and purified with nickel-agarose beads as reported previously [23].

## *In vitro* tRNase $Z^L$ cleavage assay

*In vitro* tRNase $Z^L$ cleavage assay was performed at 37°C with 1 pmol of 5′-FAM-labeled target RNA, 1 pmol of sgRNA and 8 pmol of recombinant human tRNase $Z^L$ in a 20-μl mixture containing 10 mM Tris–HCl (pH 7.1) and 10 mM $MgCl_2$. Reaction products were resolved on a 12.5% polyacrylamide gel containing 8 M urea and analyzed using a Gel Doc EX imager (Bio-Rad, Tokyo, Japan).

## 3′-RACE

Total RNA samples extracted from A549 cells treated for 96 hr with sgRNAs sgRM1, sgRM2 and sgRM6 were treated with bacterial alkaline phosphatase (Takara), and 3′-RACE-linker was added to the 3′ end of RNA using T4 RNA Ligase (Takara). These RNA samples were subjected to reverse transcription with primer-RT, and the resulting cDNAs were PCR-amplified by 40 cycles using the primer pairs FWD11/primer-RT, FWD21/primer-RT and FWD61/primer-RT for the samples from sgRM1-, sgRM2- and sgRM6-treated cells, respectively.

PCR products with an appropriate length were obtained only from the samples from sgRM1- and sgRM6-treated cells. The products were agarose gel-purified and PCR-amplified again by 40 cycles using the same primer pairs FWD11/primer-RT and FWD61/primer-RT. The re-amplified products were agarose gel-purified and PCR-amplified by 40 cycles using the nested primer pairs FWD12/primer-RT and FWD62/primer-RT. The products with an appropriate length were gel-purified and PCR-amplified by 40 cycles using the further nested primer pairs FWD13/primer-RT and FWD63/primer-RT. The final products with an appropriate length were gel-purified, digested with NotI/XhoI, and cloned into the plasmid vector pCI-neo. Five to ten cloned plasmids were purified and plasmids containing an insert sequence were subjected to DNA sequencing (Fasmac).

## Fluorescence microscopy

A549 cells were prepared in a 35-mm glass-bottom dish (IWAKI, Tokyo, Japan). Forty-eight hr after transfection with 14-nt Alexa568-labeled sgRTST, the cells were fixed with 4% paraformaldehyde in phosphate-buffered saline (PBS) for 15 min, permeabilized with 0.5% Triton X-100 in PBS for 15 min, and incubated with primary antibodies against a human tRNase $Z^L$ peptide (amino acid 812–826) for 1 hour. Subsequently, the cells were incubated with an Alexa488-conjugated secondary antibody (COSMO BIO) for 1 hr. Hoechst 33342 (Dojindo, Kumamoto, Japan) was used for nuclear staining. Fluorescence imaging was performed using an All-in-One Fluorescence Microscope BZ-X800 (KEYENCE, Osaka, Japan).

## Western blotting

A549 cells were lysed with a buffer containing 50 mM Tris-HCl (pH 7.5), 1% Triton-X100, 150 mM NaCl, and 5 mM EDTA, and the lysates were separated on an SDS/7.5% polyacrylamide gel, and transferred to a nitrocellulose membrane. tRNase Z$^L$ and β-actin on the membrane were detected by the rabbit polyclonal antibodies raised against the human tRNase Z$^L$ peptide (amino acid 812–826) and a mouse monoclonal antibody against human β-actin (Sigma-Aldrich, Tokyo, Japan), respectively, together with an anti-rabbit IgG HRP conjugated secondary antibody and an anti-mouse IgG HRP conjugated (R&D systems, Minneapolis, USA) using a LAS-3000 (FUJIFILM, Tokyo, Japan).

## Cell viability assay

A549 cells were plated at a density of 80,000 cells per well in 500 μL of medium on a 24-well dish, and transfected with 200 nM of sgRNA. After 96 hours of culture, cell viability was measured using the Cell Counting Kit-8 (Dojindo).

## Statistical analysis

Statistical significance of difference between control (mock) and experimental (sgRNA) groups was evaluated by one-tailed Student's t-test for paired samples using Microsoft Excel.

## Results

### Six 14-nt linear-type sgRNAs that target the human MALAT1 RNA

We would have a better chance to suppress the MALAT1 RNA level by TRUE gene silencing if we can select its target sites in less folded regions. However, since we do not know accurate tertiary structures of the MALAT1 RNA in human cells, we chose three sites that have been reported to be good target sites for ASOs. The 14-nt linear-type sgRNAs for the MALAT1 RNA sgRM1 and sgRM2 (Fig 1) were designed at the sites where two ASOs have been demonstrated to effectively suppress the MALAT1 RNA level in human EBC-1 cells [19]. And the 14-nt linear-type sgRNA sgRM3 (Fig 1) was designed at the site where an ASO has been shown to effectively reduce the MALAT1 level in human myeloma cells [20]. In addition, three 14-nt linear-type sgRNAs sgRM4 − sgRM6 (Fig 1) were designed at arbitrarily selected sites, and two negative control sgRNAs, sgNC1 and sgNC2, were also designed. These sgRNAs were chemically synthesized as 5′-/3′-phosphorylated and fully 2′-O-methylated oligonucleotides with full phosphorothioate linkages. Their GC content and Tm ranged from 36 to 100% and from 38 to 56°C, respectively, and only sgRM6 was predicted to form a secondary structure of ΔG = −3.9 kcal/mol (S2 Fig.). As shown below, the GC content and Tm did not appear to correlate with the efficacy of sgRNA.

### Suppression of the MALAT1 RNA level by 14-nt sgRNAs

We examined the six sgRNAs for their effect on the MALAT1 RNA level in human A549 cells. The cells were transfected with 200 nM of each sgRNA and cultured for 24 − 96 hr, and the MALAT1 RNA levels in the cells were quantified by qRT-PCR with a primer pair (pp0), between which the expected cleavage sites for those sgRNAs were not included. While all the sgRNAs showed little effect on the MALAT1 RNA level at 24 and 48 hr, its amounts were reduced at 96 hr by sgRM1, sgRM2 and sgRM6–42, 68 and 68%, respectively (Fig 2A). The negative control sgRNAs sgNC1 and sgNC2 showed no reduction in the MALAT1 level at 96 hr (S3 Fig.)

We also re-analyzed the MALAT1 level in the presence of these effective sgRNAs by qRT-PCR with three more primer pairs, pp1, pp2 and pp6, which were designed to span the expected tRNase Z$^L$ cleavage sites by sgRM1, sgRM2 and sgRM6, respectively (Fig 1A). The MALAT1 RNA levels were decreased by sgRM1, sgRM2 and sgRM6–12 − 15, 10 − 14 and 41 − 47%, respectively, and the difference in the MALAT1 level was trivial among the values measured with four different primer pairs in each case (S4 Fig). This observation suggests that once MALAT1 RNA is cleaved by tRNase Z$^L$ guided by sgRM1, sgRM2 or sgRM6, its cleavage products are rapidly degraded in the cells.

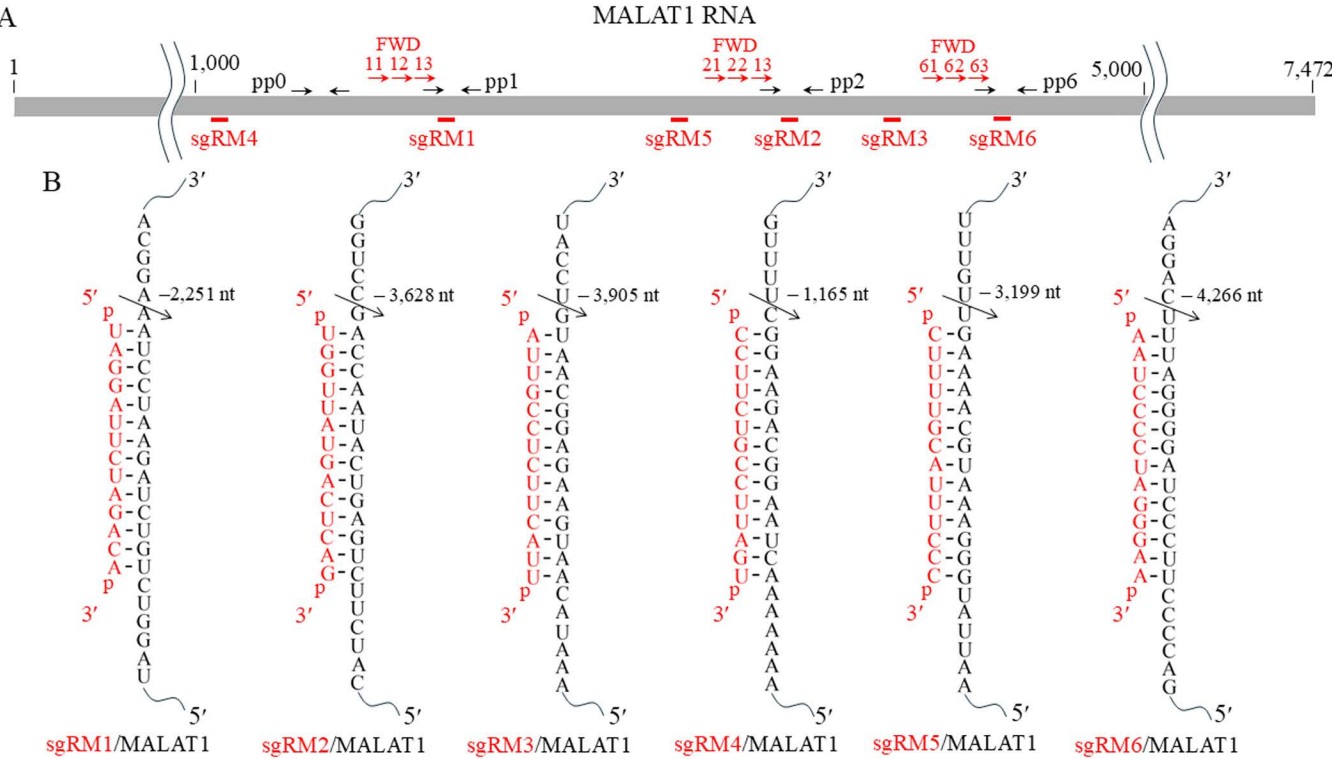

**Fig 1. Six 14-nt sgRNAs and their target, human MALAT1 RNA.** (A) Rough binding positions of the sgRNAs sgRM1 − sgRM6 on the MALAT1 RNA are shown together with rough positions of primer pairs for qRT-PCR, pp0, pp1, pp2 and pp6 and those of forward primers for 3′-RACE, FWD11 − FWD13, FWD21 − FWD23 and FWD61 − FWD63. (B) Sequences of the target/sgRNA complexes are shown. The expected cleavage sites of target/sgRNA pairs are indicated by arrows.

With respect to sgRM1, sgRM2 and sgRM6, we analyzed dose dependency of their effect. The analysis for 50, 200 and 500 nM of sgRNA was carried out at 72 and 96 hr. Dose dependency of sgRM1's effect was clearly shown, whereas the effect of sgRM2 and sgRM6 seemed to be saturated over 200 nM (Fig 2B).

We also examined the sgRNAs sgRM1 − sgRM6 for their effect on the MALAT1 RNA level in HEK293 cells. In a similar fashion to that in A549 cells, only sgRM1, sgRM2 and sgRM6 reduced the MALAT1 level at 96 hr (S5 Fig.).

## The 14-nt sgRNAs can guide tRNase Z$^L$ to cleave the MALAT1 RNA

In order to demonstrate that the suppression effect of sgRM1, sgRM2 and sgRM6 on the MALAT1 RNA level is caused by TRUE gene silencing, we performed *in vitro* tRNase Z$^L$ cleavage assay, microscopic analysis for nuclear existence of sgRNA, and tRNase Z$^L$ knockdown experiment.

For the *in vitro* tRNase Z$^L$ cleavage assay, we prepared three 30-nt MALAT1 RNA fragments, TM1, TM2 and TM6, which were 5′-FAM labeled RNA targets for sgRM1, sgRM2 and sgRM6, respectively (Fig 3). These 14-nt sgRNAs were tested for their ability to guide recombinant tRNase Z$^L$ to cleave their MALAT1 targets. The cleavage reaction was carried out at 37°C for 15, 30 or 60 min, and the percent cleavage was graphed (Fig 3 and S6 Fig.). All of the sgRNAs guided tRNase Z$^L$ *in vitro* to cleave their own targets, although the cleavage efficiency changed depending on target/sgRNA pairs. We also confirmed that the cleavages are caused by the enzymatic activity of tRNase Z$^L$ (S7A Fig.) and that target RNA cleavage by sgRNA-guided tRNase Z$^L$ is specific to the sgRNA sequence by showing that the target TM1 was cleaved under the guide of sgRM1 not of sgRM3, sgRM4 and sgRM5 (S7B Fig.).

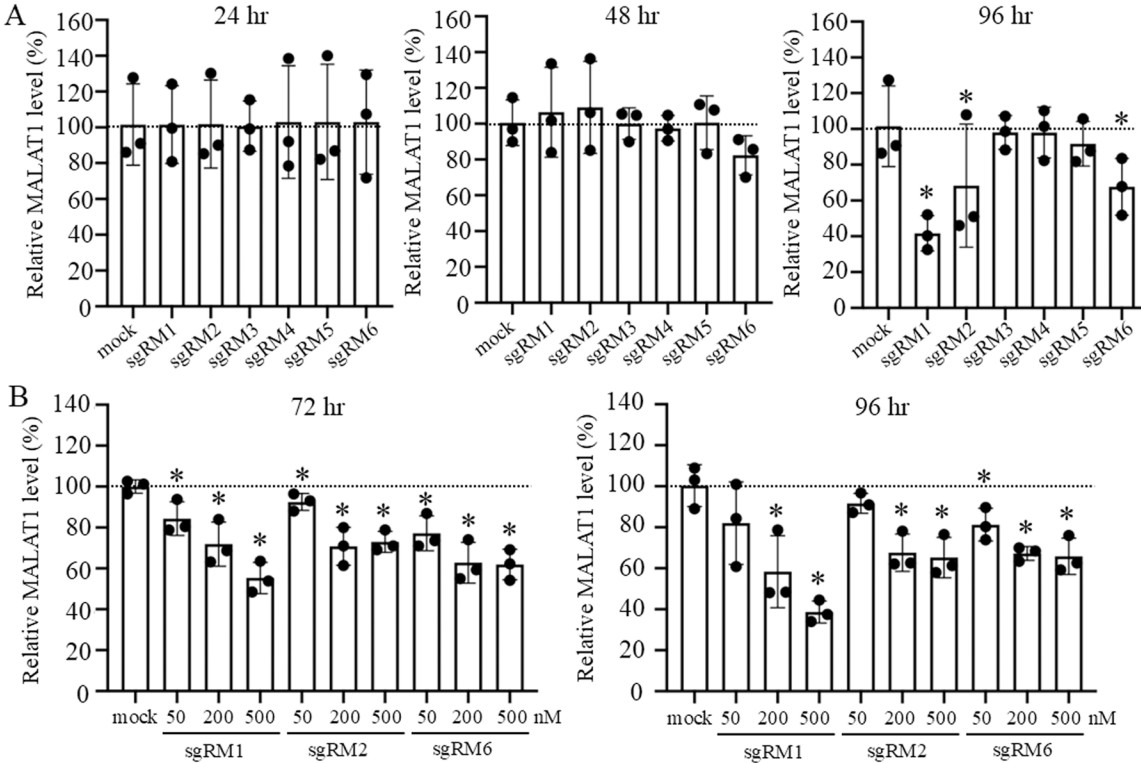

**Fig 2. Suppression of a human MALAT1 RNA level by 14-nt linear-type sgRNAs.** (A) Human A549 cells were transfected without (mock) and with each of the sgRNA sgRM1 − sgRM6 (200 nM). After 24-, 48- and 96-hr culture, a MALAT1 RNA amount was measured by qRT-PCR with a primer pair, pp0, normalized against a β-actin mRNA amount, and expressed as a percentage relative to that of untreated cells. Values are mean ± SD for three biological replicates. (B) The A549 cells were transfected without (mock) and with 50, 200 and 500 nM of each sgRNA, sgRM1, sgRM2 or sgRM6. After 72- and 96-hr culture, a MALAT1 RNA level was similarly analyzed. Values are mean ± SD for three biological replicates. Asterisk, p < 0.05.

The targets TM1 and TM2 were cleaved by the corresponding sgRNA-guided tRNase $Z^L$ at the expected sites, although the exact cleavage sites were not determined (Fig 3A,B and S6 Fig.). Likewise, the target TM6 was cleaved by sgRM6-guided tRNase $Z^L$, although the exact cleavage site was not determined (Fig 3C and S6 Fig.). The reason that the target TM6 and its 5′ cleavage product migrated on a polyacrylamide gel slightly faster than expected may be due to hairpin formation through base-pairing between a 4-nt cytosine stretch and a 4-nt guanine stretch in TM6.

Furthermore, we attempted to detect a cellular 5′ cleavage product of MALAT1 RNA generated by tRNase $Z^L$ guided by sgRM1, sgRM2 or sgRM6 by 3′-RACE, though the qRT-PCR data with four different primer sets suggested that it would be hard to detect the cleavage products because of their rapid degradation. To detect a 5′ cleavage product generated by sgRM1-guided tRNase $Z^L$, we performed RT-PCR with three nested primer pairs, and obtained an ~150-bp product from total RNA of sgRM1-treated A549 cells and ~100/250-bp products from that of mock control cells (S8A Fig.). The PCR product DNA from each RNA sample was cloned into a plasmid vector, and inserted DNA was sequenced. The sequence data showed the existence of two 5′ cleavage products ending with a uridine 62-nt and 66-nt upstream of the expected cleavage site by sgRM1-guided tRNase $Z^L$ in the sgRM1-treated cells and the existence of a 5′ cleavage product ending with a uridine 121-nt upstream of the expected cleavage site by sgRM1-guided tRNase $Z^L$ in the mock control cells (S8B Fig.). We infer that the former 5′ cleavage products were generated through further 3′ degradation of the cleavage product of MALAT1 RNA by sgRM1-guided tRNase $Z^L$, whereas the latter was from non-specific degradation.

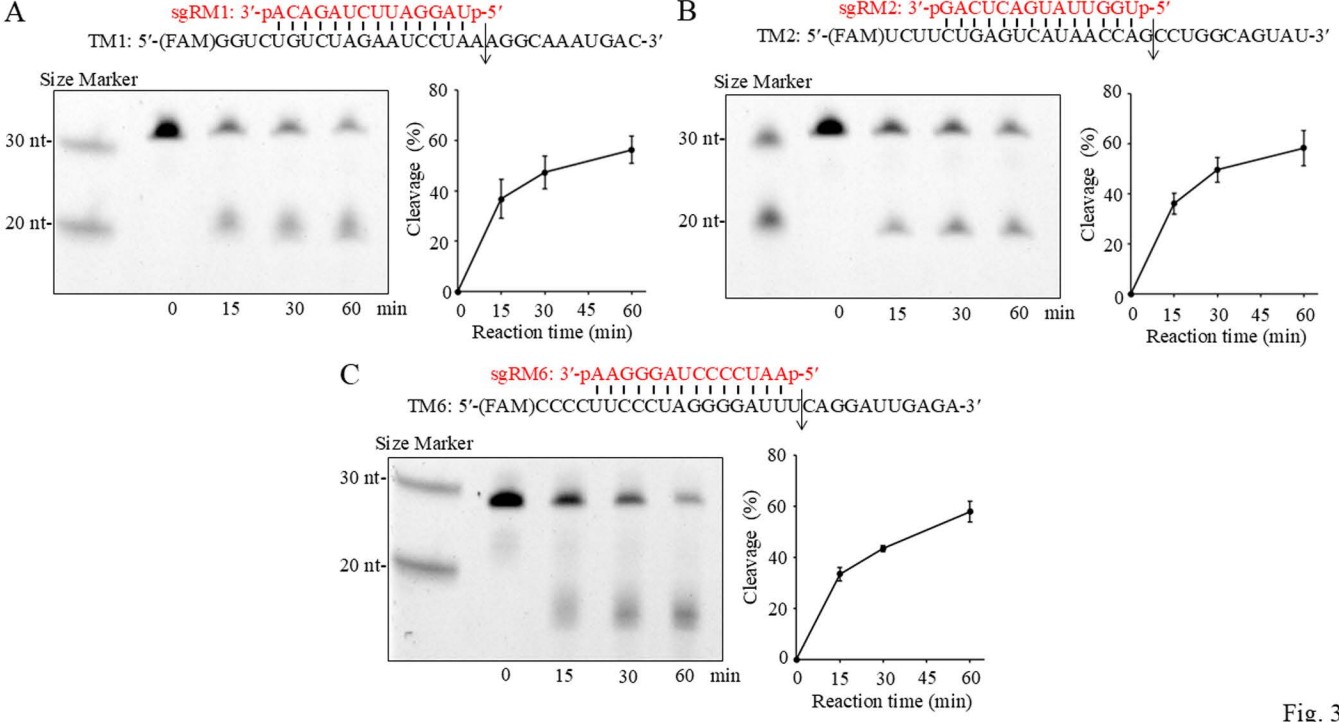

**Fig 3. *In vitro* tRNase Z^L cleavage assay.** The 30-nt 5′-FAM-labeled MALAT1 fragments TM1, TM2 and TM6 were incubated in the presence of sgRM1, sgRM2 and sgRM6, respectively, with recombinant human tRNase Z^L at 37°C for 0, 15, 30 and 60 min. A cleavage product was analyzed on a denaturing 12.5% polyacrylamide gel. Each assay was repeated three more times (S6 Fig.), and values are mean±SD for four replicates. An arrow denotes an expected cleavage site.

In the same manner, we found a 5′ cleavage product ending with a guanosine 114-nt upstream of the expected cleavage site by sgRM6-guided tRNase Z^L in the sgRM6-treated cells, but no product in the mock control cells (S8A,C Fig.). As for a 5′ cleavage product generated by sgRM2-guided tRNase Z^L, no RT-PCR product of an appropriate length was detected from total RNA of the sgRM2-treated cells and from that of the mock control cells. On the whole, the 3′-RACE data would not be inconsistent with the supposition that the 14-nt sgRNAs can guide tRNase Z^L to cleave the MALAT1 RNA also in the cells.

## A 14-nt sgRNA encapsulated into liposome can be delivered to the nucleus

Intracellular distribution of sgRNA released from liposome was analyzed with a 14-nt test sgRNA, sgRTEST, which was a 5′-Alexa568-labeled, 3′-phosphorylated and fully 2′-O-methylated oligonucleotide with full phosphorothioate linkages. sgRTEST, which was used in a different study, was repurposed since it would be likely that the RNA sequence itself hardly affects this type of analysis. The sgRNA was observed to be distributed ubiquitously in A549 cells 48 hr after transfection, with higher density in the nucleus (Fig 4 and S9 Fig.)

It has been reported that tRNase Z^L is distributed ubiquitously in various human cells, although the subcellular density distribution appears to differ depending on cellular conditions and/or analytical conditions [23–25]. In this study, tRNase Z^L in A549 cells was observed to be distributed ubiquitously with much higher density in the nucleus (Fig 4 and S9 Fig.). Considering nuclear localization of the MALAT1 RNA, the observation that sgRNA encapsulated into liposome can be delivered easily to the nucleus would provide a rationale for suppression of the MALAT1 RNA level by TRUE gene silencing.

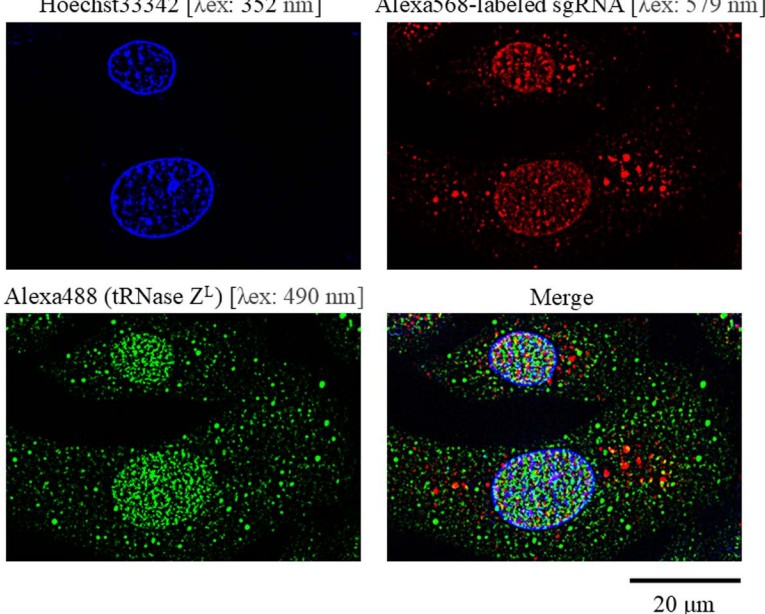

**Fig 4. Subcellular distribution of a 14-nt linear-type sgRNA and tRNase $Z^L$ in A549 cells.** A549 cells were transfected with a 14-nt 5′-Alexa568-labeled linear-type sgRNA, and after 48-hr culture, the cells were analyzed with a fluorescence microscope. tRNase $Z^L$ was visualized with an Alexa488-conjugated secondary antibody. Hoechst 33342 was used to stain the nucleus. Each image was taken by irradiating a laser beam of the indicated wave length.

## tRNase $Z^L$ is responsible for suppression of the MALAT1 RNA level

If tRNase $Z^L$ is responsible for the suppression of the MALAT1 RNA level shown in Fig 2, downregulation of tRNase $Z^L$ would attenuate the suppression effect. To examine this attenuation, we analyzed a MALAT1 RNA level in A549 cells, where a tRNase $Z^L$ amount was reduced using siRNA (Fig 5A). The MALAT1 level in the presence of sgRM1 was increased by 56% (from 27 to 42%) in the tRNase $Z^L$ knockdown cells compared with that in the normal cells (Fig 5B). This assay was repeated twice, and levels of the attenuation effect by tRNase $Z^L$ knockdown were by 51 and 28% (Fig 5C,D). Similarly, the attenuation effect by tRNase $Z^L$ knockdown was observed for the cells transfected with sgRM2 and sgRM6, in which levels of the attenuation effect were by 82−98% and by 25−38%, respectively (Fig 5B-D). Reduction of the tRNase $Z^L$ level in the cells transfected with the ineffective sgRM5 and the negative controls sgNC1 and sgNC2 showed only a trivial effect (Fig 5B-D and S10 Fig.)

## Reduction in cell viability by suppression of the MALAT1 RNA level

Since it has been reported that suppression of the MALAT1 RNA level triggers apoptosis [20], we examined the sgRNAs sgRM1, sgRM2, sgRM5, sgRM6, sgNC1 and sgNC2 for their ability to reduce cell viability. Only the effective sgRNAs sgRM1, sgRM2 and sgRM6 reduced A549 cell viability at 96 hr to 51, 58 and 67%, respectively (Fig 6 and S11 Fig.).

## Discussion

We showed that the MALAT1 RNA level in A549 cells can be suppressed by transfecting them with 14-nt linear-type sgRNA, sgRM1, sgRM2 or sgRM6, and that the suppression is caused by TRUE gene silencing by presenting three pieces of evidence: the sgRNAs sgRM1, sgRM2 and sgRM6 can guide tRNase $Z^L$ to cleave their own MALAT1 targets *in vitro*; a 14-nt linear-type sgRNA encapsulated into liposome can be delivered to the nucleus where both the target

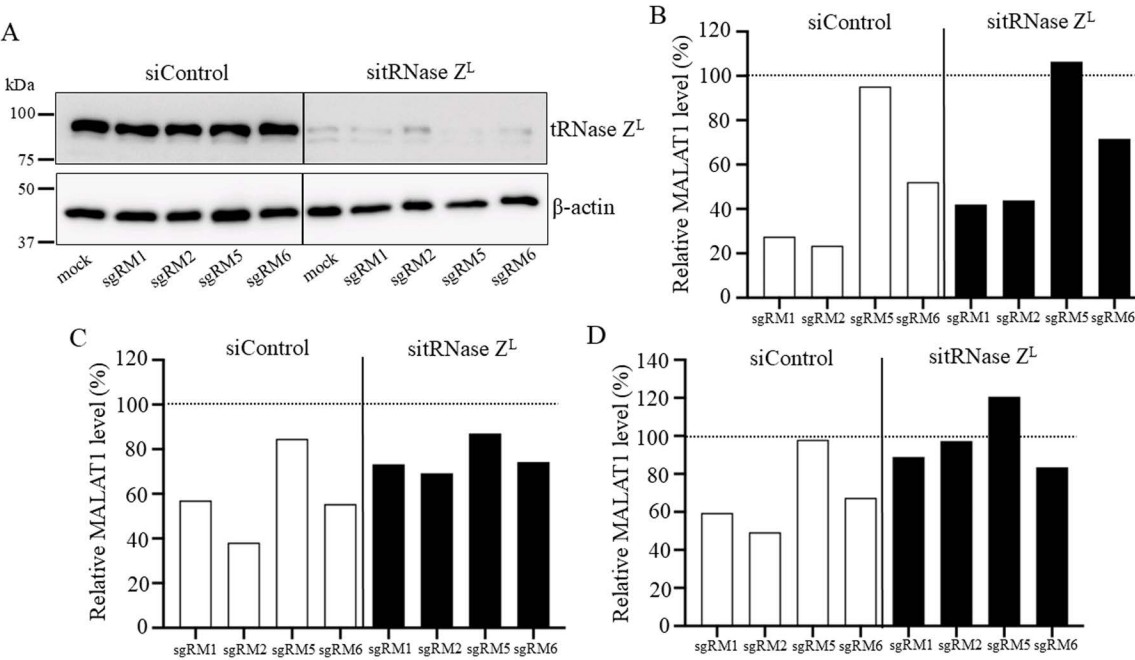

**Fig 5. Knockdown of tRNase Z$^L$ attenuates the suppression effect of sgRNA on the MALAT1 RNA level.** Twenty-four hr after A549 cells were transfected with siControl or sitRNase Z$^L$, the cells were transfected without (mock) and with each of sgRNA, sgRM1, sgRM2, sgRM5 or sgRM6 (200 nM) and cultured further. After 96-hr culture, total cellular protein and RNA were prepared. (A) tRNase Z$^L$ and β-actin protein levels were analyzed by Western blotting. (B–D) A MALAT1 RNA amount was measured by qRT-PCR, normalized against a β-actin mRNA amount, and expressed as a percentage relative to that of mock control cells. Data were from three biological replicates.

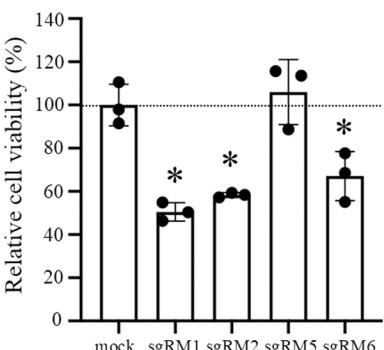

**Fig 6. Cell viability assay.** A549 cells were transfected without (mock) and with each sgRNA, sgRM1, sgRM2, sgRM5 or sgRM6 (200 nM), and after 96-hr culture, cell viability was measured. The cell viability is expressed as a percentage relative to that of untreated cells. Values are mean ± SD for three biological replicates. Asterisk, $p < 0.05$.

MALAT1 RNA and tRNase Z$^L$ exist; downregulation of tRNase Z$^L$ attenuates the suppression effect of sgRM1, sgRM2 and sgRM6 on the MALAT1 RNA level. Furthermore, we demonstrated that the effective sgRNAs sgRM1, sgRM2 and sgRM6 reduce cell viability.

Even the effective sgRNAs were hardly able to reduce the MALAT1 RNA level in 24 and 48 hr after transfection, and their suppression effect appeared only in 72 and 96 hr (Fig 2). The reason for this delay may be because tRNase Z$^L$ needed to work

primarily to produce mature tRNAs by removing 3′-trailers from pre-tRNAs in the relatively rapidly growing cells (corresponding to the cells 24 and 48 hr after transfection). In the slowly growing or quiescent cells (corresponding to the cells 72 and 94 hr after transfection), where tRNA 3′-processing would not be urgent, tRNase $Z^L$ may become available to cleave target/sgRNA complexes even though the affinity of tRNase $Z^L$ to the RNA complexes would be weaker than that to pre-tRNAs [26,27].

In contrast to sgRM1, the suppression effect of sgRM2 and sgRM6 did not show dose dependency over 200 nM (Fig 2B). Since it is well known that RNA, even relatively small one, can form various alternative conformations [28,29], the 7,472-nt long MALAT1 RNA will form many different conformations in the nucleus. The lack of dose dependency for sgRM2 and sgRM6 may be because a subset of MALAT1 RNA molecules form conformations in which binding sites of sgRM2 and/or sgRM6 are too tightly folded to access. The almost ineffective sgRNAs sgRM3, sgRM4 and sgRM5 (Fig 2A) would not be able to bind to most of the conformers of MALAT1 RNA, whereas sgRM1 may be able to bind to the MALAT1 RNA in its most of the alternative conformations. The strength of binding between sgRNA and target RNA would also need to be considered, since the ASO modified with locked nucleic acids that was designed to target the MALAT1 RNA at the same site as sgRM3 has been shown to effectively reduce the MALAT1 level in myeloma cells [20].

In spite of the overwhelming reduction in the tRNase $Z^L$ level, the sgRNAs sgRM1, sgRM2 and sgRM6 still showed their suppression effect on the MALAT1 RNA level, albeit different degree (Fig 5). The reduced amount of tRNase $Z^L$ may be sufficient to cleave a certain amount of MALAT1 RNA molecules, or binding of the sgRNAs to MALAT1 may change its stable conformations to those susceptible to nuclease degradation regardless of the presence of tRNase $Z^L$. Downregulation of tRNase $Z^L$ slightly increased the MALAT1 level that was barely suppressed by sgRM5 (Fig 5B-D), suggesting that some conformers of MALAT1 RNA can be cleaved by sgRM5-guided tRNase $Z^L$.

If we knew every conformation of a long RNA molecule in the cells, we could choose the best target site in the RNA. This might be realized in the future with the aid of artificial intelligence and/or quantum computers [30,31]. These technologies would also be used to predict off-target binding sites of sgRNAs in a transcriptome, and to avoid the off-target binding in designing sgRNA is very important especially in clinical application.

## Supporting information

**S1 Fig. Substrates of tRNase $Z^L$.** (A) Secondary structures of pre-tRNA and micro-pre-tRNA. (B) Secondary structures of sgRNA/target RNA complexes. Arrows and bars denote cleavage sites and base pairings, respectively. N, S and T represent A, U, G, or C. Sequences of S in sgRNAs form base-pairings with sequences of T in target RNAs. (PDF)

**S2 Fig. The 14-nt linear-type sgRNAs used in this study.** GC content, melting temperature (Tm), and potential secondary structure of each sgRNA are shown. (PDF)

**S3 Fig. Effect of negative control sgRNAs on a human MALAT1 RNA level.** Human A549 cells were transfected without (mock) or with sgNC1 or sgNC2 (200 nM). After 96-hr culture, a MALAT1 RNA amount was measured by qRT-PCR, normalized against a β-actin mRNA amount, and expressed as a percentage relative to that of mock control cells. Values are mean ± SD for three biological replicates. (PDF)

**S4 Fig. Suppression of a MALAT1 RNA level in A549 cells by the linear-type sgRNAs.** A549 cells were transfected without (mock) and with each of sgRM1, sgRM2 or sgRM6 (200 nM). After 96-hr culture, a MALAT1 RNA amount was measured by qRT-PCR with a primer pair, pp0, pp1, pp2 or pp6, normalized against a β-actin mRNA amount, and expressed as a percentage relative to that of mock control cells. Values are mean ± SD for three technical replicates. (PDF)

**S5 Fig. Suppression of a MALAT1 RNA level in HEK293 cells by the 14-nt linear-type sgRNAs.** Human HEK293 cells were transfected without (mock) and with each of the sgRNA sgRM1−sgRM6 (200 nM). After 48- and 96-hr culture, a MALAT1 RNA amount was measured by qRT-PCR with a primer pair, pp0, normalized against a β-actin mRNA amount, and expressed as a percentage relative to that of mock control cells. Values are mean ± SD for three biological replicates. Asterisk, $p < 0.05$.
(PDF)

**S6 Fig. *In vitro* tRNase Z$^L$ cleavage assay.** The 30-nt 5′-FAM-labeled MALAT1 fragments TM1, TM2 and TM6 were incubated in the presence of sgRM1, sgRM2 and sgRM6, respectively, with recombinant human tRNase Z$^L$ at 37°C for 0, 15, 30 and 60 min. A cleavage product was analyzed on a denaturing 12.5% polyacrylamide gel. Each assay was repeated three times.
(PDF)

**S7 Fig. Control assay for *in vitro* tRNase Z$^L$ cleavage.** (A) The 5′-FAM-labeled MALAT1 fragments TM1, TM2 and TM6 were incubated in the presence of sgRM1, sgRM2 and sgRM6, respectively, with or without recombinant human tRNase Z$^L$ at 37°C for 60 min. (B) The 5′-FAM-labeled TM1 was incubated in the presence of sgRM1, sgRM3, sgRM4 or sgRM5 with recombinant human tRNase Z$^L$ at 37°C for 60 min. A cleavage product was analyzed on a denaturing 12.5% polyacrylamide gel. I, input substrate RNA.
(PDF)

**S8 Fig. 3′-RACE experiments.** (A) PCR products obtained from the total RNA samples from mock-, sgRM1- and sgRM6-treated cells are shown on a 1.5% agarose gel. The nested primer pair FWD13/RT or FWD63/RT was used for 40-cycle PCR-amplification. (B) A partial MALAT1 sequence including the expected sgRM1-guided tRNase Z$^L$ cleavage site (denoted by a black arrow). (C) A partial MALAT1 sequence including the expected sgRM6-guided tRNase Z$^L$ cleavage site (denoted by a black arrow). Red arrow, the 3′-end of a 5′-cleavage product detected in sgRM1- or sgRM6-treated cells; broken arrow, the 3′-end of a 5′-cleavage product detected in mock-treated cells.
(PDF)

**S9 Fig. Fluorescence microscope image of mock-transfected A549 cells.** As a control experiment, A549 cells were mock-transfected. After 48-hr culture, the cells were incubated with an Alexa488-conjugated secondary antibody without incubating with primary antibodies against a human tRNase Z$^L$ peptide, and analyzed with a fluorescence microscope. Hoechst 33342 was used to stain the nucleus. Each image was taken by irradiating a laser beam of the indicated wave length.
(PDF)

**S10 Fig. tRNase Z$^L$ knockdown experiments with negative control sgRNAs.** Twenty-four hr after A549 cells were transfected with siControl or sitRNase Z$^L$, the cells were transfected without (mock) or with each of sgNC1 or sgNC2 (200 nM) and cultured further. After 96-hr culture, total cellular protein and RNA were prepared. (A) tRNase Z$^L$ and β-actin protein levels were analyzed by Western blotting. (B) A MALAT1 RNA amount was measured by qRT-PCR, normalized against a β-actin mRNA amount, and expressed as a percentage relative to that of mock control cells. Data were from three biological replicates.
(PDF)

**S11 Fig. Cell viability assay.** A549 cells were transfected without (mock) or with sgNC1 or sgNC2 (200 nM), and after 96-hr culture, cell viability was measured. The cell viability is expressed as a percentage relative to that of untreated cells. Values are mean ± SD for three biological replicates.
(PDF)

**S1 Data. All the data used in this paper.**
(XLSX)

**S1_raw_images. Original images for blots and gels.**
(PDF)

## Author contributions

**Conceptualization:** Masayuki Takahashi, Masayuki Nashimoto.

**Data curation:** Masayuki Takahashi, Masayuki Nashimoto.

**Formal analysis:** Masayuki Takahashi, Masayuki Nashimoto.

**Funding acquisition:** Masayuki Nashimoto.

**Investigation:** Masayuki Takahashi.

**Project administration:** Masayuki Nashimoto.

**Resources:** Masayuki Takahashi, Masayuki Nashimoto.

**Supervision:** Masayuki Nashimoto.

**Validation:** Masayuki Takahashi, Masayuki Nashimoto.

**Visualization:** Masayuki Takahashi.

**Writing – original draft:** Masayuki Takahashi, Masayuki Nashimoto.

**Writing – review & editing:** Masayuki Nashimoto.

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
