## [Decision Letter · Decision Letter 0]

10 Mar 2025

Dear Dr. Nashimoto,

Thank you for submitting your manuscript to PLOS ONE. After careful consideration, we feel that it has merit but does not fully meet PLOS ONE’s publication criteria as it currently stands. Therefore, we invite you to submit a revised version of the manuscript that addresses the points raised during the review process.

We look forward to receiving your revised manuscript.

Kind regards,

Hodaka Fujii, M.D., Ph.D.

Academic Editor

PLOS ONE

Journal Requirements:

Reviewers' comments:

Reviewer's Responses to Questions

**Comments to the Author**

1. Is the manuscript technically sound, and do the data support the conclusions?

Reviewer #1: Partly

Reviewer #2: Partly

Reviewer #3: Partly

Reviewer #4: No

2. Has the statistical analysis been performed appropriately and rigorously?

Reviewer #1: Yes

Reviewer #2: No

Reviewer #3: No

Reviewer #4: No

3. Have the authors made all data underlying the findings in their manuscript fully available?

Reviewer #1: Yes

Reviewer #2: No

Reviewer #3: Yes

Reviewer #4: No

4. Is the manuscript presented in an intelligible fashion and written in standard English?

Reviewer #1: Yes

Reviewer #2: Yes

Reviewer #3: Yes

Reviewer #4: Yes

Reviewer #1: The authors use 14-nt guide RNAs to recruit tRNase ZL to the MALAT1 noncoding RNA, resulting in cleavage and knockdown. 6 guides are tested, with some successfully knocking down MALAT1 to varying degrees. The authors confirm these effects require tRNase ZL. The manuscript is overall straightforward but I do have several suggestions.

(1) Introduction, First paragraph: It would be helpful to include a few additional sentences about the TRUE technology and the requirements for oligo design. This will allow a reader to better understand the technology without having to refer to other previous papers.

(2) Fig 3: Only one replicate has been shown. These experiments should be repeated 3 times to show reproducibility.

(3) Fig 5: The authors should comment on why the degree of knockdown obtained in Fig 5B is much stronger than that in 5C,D.

Minor point:

(4) MALAT1 is only 7 kb in length, not 10.5 kb as the authors claim on p.9.

Reviewer #2: The manuscript designed a 14-nt sgRNA compatible with tRNase ZL for cleavage of MALAT1 RNA, which validated the potential of TRUE silencing technology in targeting long non-coding RNAs. The authors determined the efficiency of cleavage by in vitro assay, subcellular localization and gene knock down. However, there remains several concerns:

1. The authors mentioned the nuclear localization of MALAT1 RNA in A549 cell line. However, they didn’t give any experimental evidence or literature citations.

2. Introduction section. It is mentioned that ASO has been used to KD MALAT1. Please discuss the comparison between ASO and TRUE for MALAT1 RNA knock down.

3. The authors should discuss why sgRM3 (sequence worked in ASO) was invalid in TRUE silencing.

4. Fig 3. Negative controls should be considered (non-target sgRNA, or sgRM3,4,5).

5. Where does the sequence of “5’-Alexa568 labeled sgRNA” come from? Why did the authors choose this one for tracing but not the sequence of sgRM1~6, which seems more direct? Will sgRM1~6 exhibit same localizations with this one?

6. Fig 4. Add more controls to exclude the possibility of non-specific signals.

7. Fig 5 legend. Does the term “untreated cells” mean the mock control? If so, please maintain the consistent address throughout the text. (write “mock control” directly)

8. Fig 5B~D. It’s still possible that the phenomenon was caused by indirect effect of impaired tRNA processing. Exclude this possibility experimentally (for example, transfect scrambled sgRM sequence as negative control).

9. Introduction section. The sgRNAs contain three modifications (as described in Materials and Methods), but the authors just mentioned two here in Introduction (lost “six phosphorylation modifications at the 5' and 3' ends”). Please supplement this information to avoid any misunderstanding by readers.

10. In the Materials and Methods section, "and a 5’-Alexa568-labeled, 3’-phosphorylated, fully 2’-O-methylated sgRNA with full phosphorothioate linkages (5’-UUGGUACCUUCUCA-3’)" lacks a name and description of its purpose, making it difficult to understand. It should be given an abbreviation like other sgRNAs.

11. Please add full names of abbreviations such as “FAM” when they are first mentioned.

12. "The targets TM1 and TM2 were cleaved by the corresponding sgRNA-guided tRNase ZL at the expected sites, although the exact cleavage sites were not determined (Fig 3A, B). Likewise, the target TM6 was cleaved by sgRM6-guided tRNase ZL (Fig 3C)." It’s confusing. To my understanding, the exact cleavage sites of TM1, TM2, TM6 all were not determined, please re-write it more clearly.

13. The term "TRUE gene silencing" is introduced without contextualizing it against existing RNA-targeting technologies (e.g., CRISPR-Cas13, ASOs). A brief comparison is needed to clarify its novelty, mechanistic distinctions, and the deletion effiency.

14. A scrambled/non-targeting sgRNA control is absent in experiments targeting MALAT1 RNA (a “mock” control is insufficient). Its inclusion is essential to confirm sequence-specific cleavage. Moreover, the authors directly transfected sgRNA but did not assess transfection efficiency (e.g., cellular uptake efficiency, sgRNA half-life in cells/serum), limiting mechanistic interpretation.

15. Only one primer pair used for qRT-PCR to quantify MALAT1 (>10,000 nt), risking biased results. The authors should design primer pairs spanning multiple regions (e.g., 5′, mid, and 3′) that are necessary to confirm transcript degradation. Moreover, RNA-seq should be performed to rule out transcriptome-wide unintended effects.

16. Fig. 2A shows only two biological replicates (left panels), and lacks error bars and statistical tests. The authors should expand to n ≥ 3 biological replicates and ensure all figure legends explicitly describe error bars and statistical methods.

17. In Fig. 3, the polyacrylamide gel shows only one short band after cleavage of the 30-nt RNA product. The authors should explain why two expected short cleavage bands are not observed. And the in vitro cleavage assay lacks a non-targeting/scrambled sgRNA control to confirm specificity.

18. While fluorescence microscopy suggests nuclear localization of sgRNA and tRNase ZL in Fig. 4, no significant co-localization is evident in the images. The authors should provide quantitative metrics (e.g., Pearson’s correlation coefficient) to substantiate spatial association.

19. Protein size markers are missing in Figure 5A (western blot). And panels B–D lack non-targeting/scrambled sgRNA controls, error bars, and statistical tests. “Rescue” experiments are required to strengthen conclusions.

20. Figure 6 lacks statistical tests and direct evidence for apoptosis pathways (e.g., caspase activation assays). The authors should perform caspase activity assays or related experiments to validate cell apoptosis.

21. There are lots of un-standard expressions throughout the manuscripts. Please check it thoroughly. Here are some examples:

- Fig 1 legend. Change "An arrow represents an expected cleavage site" to "The expected cleavage sites of target/sgRNA pairs are indicated by arrows."

- Last sentence of Materials and Methods: Change "After 96-hr culture, cell viable was measured using Cell Counting Kit-8" to "After 96 hours of culture, cell viability was measured using the Cell Counting Kit-8."

- Original sentence: "the MALAT1 RNA in the cells was quantitated by qRT-PCR." Suggested revision: "the MALAT1 RNA levels in the cells were quantified by qRT-PCR."

- “While all the sgRNAs showed little effect on the MALAT1 RNA level in 24 and 48 hr”. the word "in" should be replaced with "at" in this sentence, as well as in several other instances throughout the text.

Reviewer #3: Comments/suggestions to the authors:

Introduction

Authors should provide a more precise introduction, detailing the potential benefits of their testing gene silencing approach in comparison to the well-established RNAseH-based method.

Additionally, there is no clear rationale for choosing the 14-nt linear type of sgRNA design among the four available options.

The introduction should be rewritten to enhance clarity.

Results

The specific properties such as Tm, GC%, and potential secondary structure of the selected sgRNAs should be presented in a table or other format to facilitate the comparison of sequences.

The entire package of results is variable and lacks consistency, raising questions about the validity of the effect. In experiments like this, it is critical to have appropriate controls, which are missing in this piece of work. Single-stranded oligonucleotides at a high concentration similar to used in this manuscript can have many indirect effects. Therefore, mock controls are not sufficient; a scramble control should be used instead. This is crucial for all data presented in this manuscript.

Fig. 3 gels potentially can be stained with intercalating dye instead of FAM, allowing for control lanes with sgRNA or target sequence only. The data should include the same time points with and without the enzyme to confirm that degradation is really enzyme-dependent!

While the in vitro cleavage assay has limitations, the authors should provide 5'/3 ‘RACE assay results to demonstrate that the introduced sgRNA cleaves the target in cells.

The paper would be enhanced by demonstrating Malat1 KD effects in other cell types and proving data that this KD approach works on other targets.

Cell viability data is missing the appropriate negative control and also a positive control, which could be scramble ASO and a gapmer targeting the Malat1 transcript, respectively.

Reviewer #4: The manuscript by Takahashi and Nashimoto reports on re-examining a post-transcriptional silencing system that the last author attempts to develop ever since 1991. This system (TRUE) is based on exploiting a ubiquitous endogenous tRNA processing enzyme called RNase Z and exogenous small RNAs to target non-tRNAs. While the system has been used by the Nashimoto et al., to target blood cancers, it seems not to have been widely accepted nor been taken up for further development by others.

In this manuscript, the authors use the system to target a ‘jackpot’ long non-coding RNA called MALAT1, which is implied in pretty much everything, including to function in a variety of cancers. Using a stabilized small RNA to target RNase Z to MALAT1, the authors show that MALAT1 RNA can be reduced, which results in lower cell viability of one lung cancer-derived cell line.

The manuscript is rather simple in its message, and does not really add understanding to the mechanism or improvement of the TRUE system. Overall, the presentation of the few data points are acceptable for publication in PLoS One. However, some obvious questions remain to be answered before publication can be recommended. This reviewer therefore asks for a major revision.

General comments:

(1) While the early attempts to use RNase Z to downregulate the function of non-tRNAs predated the rise of RNAi- and miRNA-based gene regulatory systems, it is curious why the authors are still trying to improve TRUE rather than accepting that other RNA-based approaches are much more developed and more efficient than TRUE. Could the authors give the reader a rationale for using TRUE, and not siRNA-based methods when trying to interfere with MALAT1 function?

(2) 200 nM sgRNA with a size of 14 nt corresponds to 900.9 micrograms of sgRNA per liter.

For transfections in a 12-well format with 1 mL medium this would amount to 900.9 nanograms per well. A 12-well dish format should be seeded at 0.1 x 10^6 cells. Assuming transfection about 24 hours later, after at least one cell division of A549 cells, 0.2 x 10^6 cells will be transfected with 900.9 nanograms. This mass corresponds to 1.204e+14 sgRNAs. Even with a transfection efficiency of only 1%, each cell would be exposed to 6.02e+6 sgRNA molecules. Given the efficiency of MALAT1 downregulation by a maximum of 58%, one wonders if the effects of the knockdown (cell viability reduction) are really due to the loss of function of half the MALAT1 molecules, or due to poisoning of the cells by exorbitant numbers of small RNAs. This aspect of quantitative mass effects should be discussed by the authors.

(3) Given that the authors used qRT-PCR to determine the relative levels of MALAT1 RNA after sgRNA transfection, they should provide the reader with a description where exactly in the MALAT1 RNA sequence the sgRNAs are targeting and where the qRT-PCR primers would bind to the cDNA derived from MALAT1. If the PCR primers were designed to bind to cDNA regions that do not span the RNase Z cleavage site, the reader could assume/conclude that the reported knock-down efficiencies are masked by remnants of MALAT1 RNA that can still be reverse-transcribed.

(4) The authors did not adhere to the PLOS Data policy. “For example, in addition to summary statistics, the data points behind means, medians and variance measures should be available.”

No data points have been submitted with the manuscript.

Specific comments:

(1) “In order to demonstrate that the suppression effect of sgRM1, sgRM2 and sgRM6 on the MALAT1 RNA level is caused by TRUE gene silencing, we performed in vitro tRNase…”

> It is unclear why the authors resort to an in vitro cleavage experiments as proxy to show that 14 nt sgRNA targets MALAT1. Even though MALAT1 is 6.98 kb transcript, it has been detected by northern blotting, and it is necessary to perform northern blotting with probes around the sgRNA target sides to show that TRUE is targeting MALAT1 after transfection, and to complement the qRT-PCR results. Only then, can the authors claim that MALAT1 is affected by RNase Z-mediated cleavage.

(2) Figure 2A:

> Can the authors explain why transfection of some sgRNAs directed against MALAT1 increase relative MALAT1 transcript levels?

(3) Figure 3:

> To make such an experiment more scientifically sound, could the authors incorporate more cleavage efficiencies that would add to the quantification shown to the right of each PAGE gel? As of now, it looks like this experiment was performed only once, which is not sufficient to accommodate to the scope of PLoS One, a journal that aims to publish experimentally sound manuscripts.

(4) Figure 5:

“(B−D) A MALAT1 RNA amount was measured by qRT-PCR, normalized against a β-actin mRNA amount, and expressed as a percentage relative to that of untreated cells. Data were from three biological replicates.”

> Even though the authors state that they performed biological replicates, the Figure does not show that. Was every replicate output exactly the same?

**Do you want your identity to be public for this peer review?** For information about this choice, including consent withdrawal, please see our Privacy Policy

Reviewer #1: No

Reviewer #2: No

Reviewer #3: No

Reviewer #4: No

---

## [Author Response · Author response to Decision Letter 1]

20 Jun 2025

Responses to Reviewers

Reviewer #1: The authors use 14-nt guide RNAs to recruit tRNase ZL to the MALAT1 noncoding RNA, resulting in cleavage and knockdown. 6 guides are tested, with some successfully knocking down MALAT1 to varying degrees. The authors confirm these effects require tRNase ZL. The manuscript is overall straightforward but I do have several suggestions.

(1) Introduction, First paragraph: It would be helpful to include a few additional sentences about the TRUE technology and the requirements for oligo design. This will allow a reader to better understand the technology without having to refer to other previous papers.

We have modified the Introduction section (page 3, line 9).

(2) Fig 3: Only one replicate has been shown. These experiments should be repeated 3 times to show reproducibility.

We have repeated the cleavage assays three more times and added the data (Fig 3 and S6 Fig.).

(3) Fig 5: The authors should comment on why the degree of knockdown obtained in Fig 5B is much stronger than that in 5C,D.

We are not sure but we assume that the MALAT1 expression level and/or the uptake efficiency of sgRNA could change depending on preparation of the cells. We observed similar variation in biological replicates in Fig 2.

Minor point:

(4) MALAT1 is only 7 kb in length, not 10.5 kb as the authors claim on p.9.

We have corrected the length of MALAT1 to 7,472-nt according to the latest information (https://www.ncbi.nlm.nih.gov/nuccore/NR_002819.5) (page 12, line 16).

Reviewer #2: The manuscript designed a 14-nt sgRNA compatible with tRNase ZL for cleavage of MALAT1 RNA, which validated the potential of TRUE silencing technology in targeting long non-coding RNAs. The authors determined the efficiency of cleavage by in vitro assay, subcellular localization and gene knock down. However, there remains several concerns:

1. The authors mentioned the nuclear localization of MALAT1 RNA in A549 cell line. However, they didn’t give any experimental evidence or literature citations.

Although we do not know experimental evidence for the nuclear localization of MALAT1 RNA in A549 cells, it has been reported that the MALAT1 RNA is localized in the nucleus (ref. 18) (page 4, line 5).

2. Introduction section. It is mentioned that ASO has been used to KD MALAT1. Please discuss the comparison between ASO and TRUE for MALAT1 RNA knock down.

We have modified the Introduction section (page 4, line 9).

3. The authors should discuss why sgRM3 (sequence worked in ASO) was invalid in TRUE silencing.

We have discussed this issue in page 12 (line 14 from the bottom).

4. Fig 3. Negative controls should be considered (non-target sgRNA, or sgRM3,4,5).

We have carried out the cleavage assays with negative control sgRNAs and added the data (S7B Fig.) (page 9, line 8 from the bottom).

5. Where does the sequence of “5’-Alexa568 labeled sgRNA” come from? Why did the authors choose this one for tracing but not the sequence of sgRM1~6, which seems more direct? Will sgRM1~6 exhibit same localizations with this one?

We have mentioned the origin of this sgRNA in page 10 (line 5 from the bottom) as follows: “sgRTEST, which was used in a different study, was repurposed since it would be likely that the RNA sequence itself hardly affects this type of analysis.”

6. Fig 4. Add more controls to exclude the possibility of non-specific signals.

We have added control data (S9 Fig.).

7. Fig 5 legend. Does the term “untreated cells” mean the mock control? If so, please maintain the consistent address throughout the text. (write “mock control” directly)

We have rephrased as suggested (page 19, line 5).

8. Fig 5B~D. It’s still possible that the phenomenon was caused by indirect effect of impaired tRNA processing. Exclude this possibility experimentally (for example, transfect scrambled sgRM sequence as negative control).

We have performed experiments with two negative control sgRNAs (S10 Fig.) (page 11, line 20).

9. Introduction section. The sgRNAs contain three modifications (as described in Materials and Methods), but the authors just mentioned two here in Introduction (lost “six phosphorylation modifications at the 5' and 3' ends”). Please supplement this information to avoid any misunderstanding by readers.

We have mentioned 5′-/3′-phosphorylation (page 4, line 1).

10. In the Materials and Methods section, "and a 5’-Alexa568-labeled, 3’-phosphorylated, fully 2’-O-methylated sgRNA with full phosphorothioate linkages (5’-UUGGUACCUUCUCA-3’)" lacks a name and description of its purpose, making it difficult to understand. It should be given an abbreviation like other sgRNAs.

We have named this sgRNA sgRTEST (page 4, line 13 from the bottom).

11. Please add full names of abbreviations such as “FAM” when they are first mentioned.

We have added the full name of “FAM” (page 5, line 16).

12. "The targets TM1 and TM2 were cleaved by the corresponding sgRNA-guided tRNase ZL at the expected sites, although the exact cleavage sites were not determined (Fig 3A, B). Likewise, the target TM6 was cleaved by sgRM6-guided tRNase ZL (Fig 3C)." It’s confusing. To my understanding, the exact cleavage sites of TM1, TM2, TM6 all were not determined, please re-write it more clearly.

We have modified the sentence (page 9, line 2 from the bottom).

13. The term "TRUE gene silencing" is introduced without contextualizing it against existing RNA-targeting technologies (e.g., CRISPR-Cas13, ASOs). A brief comparison is needed to clarify its novelty, mechanistic distinctions, and the deletion effiency.

We have modified the Introduction section (page 3, line 16).

14. A scrambled/non-targeting sgRNA control is absent in experiments targeting MALAT1 RNA (a “mock” control is insufficient). Its inclusion is essential to confirm sequence-specific cleavage. Moreover, the authors directly transfected sgRNA but did not assess transfection efficiency (e.g., cellular uptake efficiency, sgRNA half-life in cells/serum), limiting mechanistic interpretation.

We have included experimental data with non-targeting control sgRNAs (S3 Fig., S10 Fig., S11 Fig.). We understand the importance of assessing transfection efficiency, and will analyze it in the future study.

15. Only one primer pair used for qRT-PCR to quantify MALAT1 (>10,000 nt), risking biased results. The authors should design primer pairs spanning multiple regions (e.g., 5′, mid, and 3′) that are necessary to confirm transcript degradation. Moreover, RNA-seq should be performed to rule out transcriptome-wide unintended effects.

We have performed qRT-PCR with three more primer sets to quantify MALAT1 and presented the data (S4 Fig.) (page 8, line 2 from the bottom). We understand the importance of transcriptome analysis, and will carry it out in the future.

16. Fig. 2A shows only two biological replicates (left panels), and lacks error bars and statistical tests. The authors should expand to n ≥ 3 biological replicates and ensure all figure legends explicitly describe error bars and statistical methods.

We have performed additional experiments and modified the figure legend (page 18, line 12).

17. In Fig. 3, the polyacrylamide gel shows only one short band after cleavage of the 30-nt RNA product. The authors should explain why two expected short cleavage bands are not observed. And the in vitro cleavage assay lacks a non-targeting/scrambled sgRNA control to confirm specificity.

The reason why only one short band after cleavage of the 30-nt target RNA appeared is that since FAM is attached only to the 5′-end of the 30-nt target RNA, the 3′ cleavage product is invisible.

We have carried out the cleavage assays with negative control sgRNAs and added the data (S7B Fig.) (page 9, line 8 from the bottom).

18. While fluorescence microscopy suggests nuclear localization of sgRNA and tRNase ZL in Fig. 4, no significant co-localization is evident in the images. The authors should provide quantitative metrics (e.g., Pearson’s correlation coefficient) to substantiate spatial association.

We think that the reason why no significant co-localization is evident in the images is that tRNase ZL binds to sgRNA only in the form of a target RNA/sgRNA complex and that the binding should be only transient because of an enzyme/substrate interaction. Incidentally, the Pearson’s correlation coefficient of the nuclear regions calculated with Image J software was around − 0.06.

19. Protein size markers are missing in Figure 5A (western blot). And panels B–D lack non-targeting/scrambled sgRNA controls, error bars, and statistical tests. “Rescue” experiments are required to strengthen conclusions.

We have added protein size markers (Fig. 5A) and data with non-target control sgRNAs (S10 Fig.) (page 11, line 14 from the bottom).

Because, from our own experience, tRNase ZL expression from exogenous plasmids appears to be harmful to cells, we did not do “rescue” experiments.

20. Figure 6 lacks statistical tests and direct evidence for apoptosis pathways (e.g., caspase activation assays). The authors should perform caspase activity assays or related experiments to validate cell apoptosis.

We have added statistical test data (Fig 6). Since suppression of the MALAT1 RNA level has been reported to trigger apoptosis (ref. 20), we checked only cell viability. We understand that caspase activity assays are needed to show that reduction in the cell viability is caused by apoptosis, but, in the current study, we only confirmed that the sgRNAs sgRM1, sgRM2 and sgRM6 reduce A549 cell viability.

21. There are lots of un-standard expressions throughout the manuscripts. Please check it thoroughly. Here are some examples:

- Fig 1 legend. Change "An arrow represents an expected cleavage site" to "The expected cleavage sites of target/sgRNA pairs are indicated by arrows."

- Last sentence of Materials and Methods: Change "After 96-hr culture, cell viable was measured using Cell Counting Kit-8" to "After 96 hours of culture, cell viability was measured using the Cell Counting Kit-8."

- Original sentence: "the MALAT1 RNA in the cells was quantitated by qRT-PCR." Suggested revision: "the MALAT1 RNA levels in the cells were quantified by qRT-PCR."

- “While all the sgRNAs showed little effect on the MALAT1 RNA level in 24 and 48 hr”. the word "in" should be replaced with "at" in this sentence, as well as in several other instances throughout the text.

We have rephrased as suggested (page 18, line 6; page 7, line 2 from the bottom; page 8, line 8 from the bottom).

Reviewer #3: Comments/suggestions to the authors:

Introduction

Authors should provide a more precise introduction, detailing the potential benefits of their testing gene silencing approach in comparison to the well-established RNAseH-based method. Additionally, there is no clear rationale for choosing the 14-nt linear type of sgRNA design among the four available options. The introduction should be rewritten to enhance clarity.

We have modified the Introduction section.

Results

The specific properties such as Tm, GC%, and potential secondary structure of the selected sgRNAs should be presented in a table or other format to facilitate the comparison of sequences.

We have added the above properties of sgRNAs in S2 Fig. (page 8, line 15 from the bottom).

The entire package of results is variable and lacks consistency, raising questions about the validity of the effect. In experiments like this, it is critical to have appropriate controls, which are missing in this piece of work. Single-stranded oligonucleotides at a high concentration similar to used in this manuscript can have many indirect effects. Therefore, mock controls are not sufficient; a scramble control should be used instead. This is crucial for all data presented in this manuscript.

We have performed experiments with two negative control sgRNAs and added data (S3 Fig., S10 Fig., S11 Fig.).

Fig. 3 gels potentially can be stained with intercalating dye instead of FAM, allowing for control lanes with sgRNA or target sequence only. The data should include the same time points with and without the enzyme to confirm that degradation is really enzyme-dependent!

We have carried out the assays without tRNase ZL and added the data (S7A Fig.) (page 9, line 9 from the bottom).

While the in vitro cleavage assay has limitations, the authors should provide 5'/3 ‘RACE assay results to demonstrate that the introduced sgRNA cleaves the target in cells.

We have performed 3′-RACE experiments and presented the data (S8 Fig.) (page 10, line 3).

The paper would be enhanced by demonstrating Malat1 KD effects in other cell types and proving data that this KD approach works on other targets.

We have added data that show suppression of the MALAT1 level in HEK293 cells (S5 Fig.) (page 9, line 12).

Cell viability data is missing the appropriate negative control and also a positive control, which could be scramble ASO and a gapmer targeting the Malat1 transcript, respectively.

We have performed cell viability assays with two negative control sgRNAs (S11 Fig.). As we have explained in Introduction (page 4, line 9), in this paper, we focused on a study of sgRNA for TRUE gene silencing without comparing with ASO.

Reviewer #4: The manuscript by Takahashi and Nashimoto reports on re-examining a post-transcriptional silencing system that the last author attempts to develop ever since 1991. This system (TRUE) is based on exploiting a ubiquitous endogenous tRNA processing enzyme called RNase Z and exogenous small RNAs to target non-tRNAs. While the system has been used by the Nashimoto et al., to target blood cancers, it seems not to have been widely accepted nor been taken up for further development by others. In this manuscript, the authors use the system to target a ‘jackpot’ long non-coding RNA called MALAT1, which is implied in pretty much everything, including to function in a variety of cancers. Using a stabilized small RNA to target RNase Z to MALAT1, the authors show that MALAT1 RNA can be reduced, which results in lower cell viability of one lung cancer-derived cell line. The manuscript is rather simple in its message, and does not really add understanding to the mechanism or improvement of the TRUE system. Overall, the presentation of the few data points are acceptable for publication in PLoS One. However, some obvious questions remain to be answered before publication can be recommended. This reviewer therefore asks for a major revision.

General comments:

(1) While the early attempts to use RNase Z to downregulate the function of non-tRNAs predated the rise of RNAi- and miRNA-based gene regulatory systems, it is curious why the authors are still trying to improve TRUE rather than accepting that other RNA-based approaches are much more developed and more efficient than TRUE. Could the authors give the reader a rationale for using TRUE, and not siRNA-based methods when trying to interfere with MALAT1 function?

We have modified the Introduction section.

(2) 200 nM sgRNA with a size of 14 nt corresponds to 900.9 micrograms of sgRNA per liter. For transfections in a 12-well format with 1 mL medium this would amount to 900.9 nanograms per well. A 12-well dish format should be seeded at 0.1 x 10^6 cells. Assuming transfection about 24 hours later, after at least one cell division of A549 cells, 0.2 x 10^6 cells will be transfected with 900.9 nanograms. This mass corresponds to 1.204e+14 sgRNAs. Even with a transfection efficiency of only 1%, each cell would be exposed to 6.02e+6 sgRNA molecules. Given the efficiency of MALAT1 downregulation by a maximum of 58%, one wonders if the effects of the knockdown (cell viability reduction) are really due to the loss of function of half the MALAT1 molecules, or due to poisoning of the cells by exorbitant numbers of small RNAs. This aspect of quantitative mass effects should be discussed by the authors.

We understand that 200 nM of sgRNA could be deleterious to cells for some reason. To exclude this possibility, we have performed cell viability assays with non-targeting control sgRNAs (S11 Fig.).

(3) Given that the authors used qRT-PCR to determine the relative levels of MALAT1 RNA after sgRNA transfection, they should provide the reader with

---

## [Decision Letter · Decision Letter 1]

9 Jul 2025

Dear Dr. Nashimoto,

We look forward to receiving your revised manuscript.

Kind regards,

Hodaka Fujii, M.D., Ph.D.

Academic Editor

PLOS ONE

Journal Requirements:

Reviewers' comments:

Reviewer's Responses to Questions

**Comments to the Author**

Reviewer #1: All comments have been addressed

Reviewer #2: All comments have been addressed

Reviewer #4: All comments have been addressed

2. Is the manuscript technically sound, and do the data support the conclusions?

Reviewer #1: Yes

Reviewer #2: Yes

Reviewer #4: Yes

3. Has the statistical analysis been performed appropriately and rigorously?

Reviewer #1: Yes

Reviewer #2: Yes

Reviewer #4: Yes

4. Have the authors made all data underlying the findings in their manuscript fully available?

Reviewer #1: Yes

Reviewer #2: Yes

Reviewer #4: Yes

5. Is the manuscript presented in an intelligible fashion and written in standard English?

Reviewer #1: Yes

Reviewer #2: Yes

Reviewer #4: Yes

Reviewer #1: The authors have appropriately revised the manuscript and addressed the prior comments from each of the reviewers.

Reviewer #2: The authors have addressed almost all my concerns, although the RNA-seq experiments have not been performed.

Reviewer #4: The authors have addressed some but not all of the comments by this reviewer. However, the authors performed some experiments that can be accepted as substitutes for the requested approaches. This reviewer recommends publication after minor revision.

General comments:

(1) While the early attempts to use RNase Z to downregulate the function of non-tRNAs predated the rise of RNAi- and miRNA-based gene regulatory systems, it is curious why the authors are still trying to improve TRUE rather than accepting that other RNA-based approaches are much more developed and more efficient than TRUE. Could the authors give the reader a rationale for using TRUE, and not siRNA-based methods when trying to interfere with MALAT1 function?

>> The authors have added the following statement:

“Under such circumstances, one may think that clinical applications of TRUE gene silencing would not be worth pursuing. However, we believe that we would have a chance to find a niche of this technology, since there would be a large number of potential target RNAs that cause various diseases and a subset of them may not be able to be eliminated easily by the above established and emerging technologies.”

>> The reviewer understands that the authors are mixing the world of belief with the world of science, the latter of which should be disconnected from belief as much as possible when performing experiments to understand the natural world.

(2) 200 nM sgRNA with a size of 14 nt corresponds to 900.9 micrograms of sgRNA per liter.

For transfections in a 12-well format with 1 mL medium this would amount to 900.9 nanograms per well. A 12-well dish format should be seeded at 0.1 x 10^6 cells. Assuming transfection about 24 hours later, after at least one cell division of A549 cells, 0.2 x 10^6 cells will be transfected with 900.9 nanograms. This mass corresponds to 1.204e+14 sgRNAs. Even with a transfection efficiency of only 1%, each cell would be exposed to 6.02e+6 sgRNA molecules. Given the efficiency of MALAT1 downregulation by a maximum of 58%, one wonders if the effects of the knockdown (cell viability reduction) are really due to the loss of function of half the MALAT1 molecules, or due to poisoning of the cells by exorbitant numbers of small RNAs. This aspect of quantitative mass effects should be discussed by the authors.

>> The authors have added a Figure that shows that exorbitant amounts of non-targeting RNAs increase rather than decrease cell viability. While this reviewer will accept this experiment, the results are contrasting years of research into the poisonous effects of large copy numbers of siRNAs when introduced into human cells.

(3) Given that the authors used qRT-PCR to determine the relative levels of MALAT1 RNA after sgRNA transfection, they should provide the reader with a description where exactly in the MALAT1 RNA sequence the sgRNAs are targeting and where the qRT-PCR primers would bind to the cDNA derived from MALAT1. If the PCR primers were designed to bind to cDNA regions that do not span the RNase Z cleavage site, the reader could assume/conclude that the reported knock-down efficiencies are masked by remnants of MALAT1 RNA that can still be reverse-transcribed.

>> The authors have addressed this point to completion.

(4) The authors did not adhere to the PLOS Data policy. “For example, in addition to summary statistics, the data points behind means, medians and variance measures should be available.”

No data points have been submitted with the manuscript.

>> The authors have addressed this point to completion.

Specific comments:

(1) “In order to demonstrate that the suppression effect of sgRM1, sgRM2 and sgRM6 on the MALAT1 RNA level is caused by TRUE gene silencing, we performed in vitro tRNase…”

> It is unclear why the authors resort to an in vitro cleavage experiments as proxy to show that 14 nt sgRNA targets MALAT1. Even though MALAT1 is 6.98 kb transcript, it has been detected by northern blotting, and it is necessary to perform northern blotting with probes around the sgRNA target sides to show that TRUE is targeting MALAT1 after transfection, and to complement the qRT-PCR results. Only then, can the authors claim that MALAT1 is affected by RNase Z-mediated cleavage.

>> The authors have addressed this point by performing some complicated 3’ RACE experiment. They did not perform a much simpler northern blot. The results presented in the new Figures are summarized in a complicated statement: “On the whole, the 3′-RACE data would not be inconsistent with the supposition that the 14-nt sgRNAs can guide tRNase ZL to cleave the MALAT1 RNA also in the cells.” This reviewer leaves the conclusion about the in vivo cleavage efficiency to the future readers of this manuscript.

(2) Figure 2A:

> Can the authors explain why transfection of some sgRNAs directed against MALAT1 increase relative MALAT1 transcript levels?

>> The authors give a handwaving explanation that does not really address the discrepancy in perception of these results. As it stands, sgRNAs should result in the degradation of MALAT1 by RNaseZ, but the authors suggest that some sgRNAs result in “slower degradation by disturbed nuclease activity that is caused by sgRNA binding.” This reviewer wonders about the applicability of the method if one has to screen candidate sgRNAs not only for knock-down efficiency of the target RNA, but also for potential stabilization of the target RNA, which is not what the method should achieve.

(3) Figure 3:

> To make such an experiment more scientifically sound, could the authors incorporate more cleavage efficiencies that would add to the quantification shown to the right of each PAGE gel? As of now, it looks like this experiment was performed only once, which is not sufficient to accommodate to the scope of PLoS One, a journal that aims to publish experimentally sound manuscripts.

>> The authors have addressed this point to completion.

(4) Figure 5:

“(B−D) A MALAT1 RNA amount was measured by qRT-PCR, normalized against a β-actin mRNA amount, and expressed as a percentage relative to that of untreated cells. Data were from three biological replicates.”

> Even though the authors state that they performed biological replicates, the Figure does not show that. Was every replicate output exactly the same?

>> This reviewer accepts this explanation.

NEW COMMENTS:

SFig 9:

This Figure is called in the text as: “The sgRNA was observed to be distributed ubiquitously in A549 cells 48 hr after transfection, with higher density in the nucleus (Fig 4 and S9 Fig.)”.

On the other hand, the Figure legend to SFig. 9 reads:

“Fluorescence microscope image of mock-transfected A549 cells. As a control experiment, A549 cells were mock-transfected. After 48-hr culture, the cells were incubated with an Alexa488-conjugated secondary antibody without incubating with primary antibodies against a human tRNase ZL peptide, and analyzed with a fluorescence microscope. Hoechst 33342 was used to stain the nucleus. Each image was taken by irradiating a laser beam of the indicated wave length.”

>>To this reviewer there is some discrepancy between the two statements, especially since SFig. 9 does not show only the nuclear stain and no signal in the other two emission channels.

**Do you want your identity to be public for this peer review?** For information about this choice, including consent withdrawal, please see our Privacy Policy

Reviewer #1: No

Reviewer #2: No

Reviewer #4: No

---

## [Author Response · Author response to Decision Letter 2]

11 Jul 2025

Response to Reviewer #4

The authors have addressed some but not all of the comments by this reviewer. However, the authors performed some experiments that can be accepted as substitutes for the requested approaches. This reviewer recommends publication after minor revision.

NEW COMMENTS:

SFig 9:

This Figure is called in the text as: “The sgRNA was observed to be distributed ubiquitously in A549 cells 48 hr after transfection, with higher density in the nucleus (Fig 4 and S9 Fig.)”. On the other hand, the Figure legend to SFig. 9 reads: “Fluorescence microscope image of mock-transfected A549 cells. As a control experiment, A549 cells were mock-transfected. After 48-hr culture, the cells were incubated with an Alexa488-conjugated secondary antibody without incubating with primary antibodies against a human tRNase ZL peptide, and analyzed with a fluorescence microscope. Hoechst 33342 was used to stain the nucleus. Each image was taken by irradiating a laser beam of the indicated wave length.”

>>To this reviewer there is some discrepancy between the two statements, especially since SFig. 9 does not show only the nuclear stain and no signal in the other two emission channels.

The control experiment shown in S9 Fig. has been performed to exclude the possibility of non-specific signals, which was pointed out by Reviewer #2. In this experiment, where the Alexa568-labeled sgRNA and the primary antibodies against a human tRNase ZL peptide were not added, we observed no significant signal by irradiating laser beams of 579 and 490 nm. This observation would exclude the possibility of non-specific signals.

---

## [Decision Letter · Decision Letter 2]

16 Jul 2025

Cleavage of MALAT1 RNA by 14-nt sgRNA-guided tRNase ZL

PONE-D-25-04390R2

Dear Dr. Nashimoto,

We’re pleased to inform you that your manuscript has been judged scientifically suitable for publication and will be formally accepted for publication once it meets all outstanding technical requirements.

Kind regards,

Hodaka Fujii, M.D., Ph.D.

Academic Editor

PLOS ONE

Additional Editor Comments (optional):

Reviewers' comments:

Reviewer's Responses to Questions

**Comments to the Author**

Reviewer #4: All comments have been addressed

2. Is the manuscript technically sound, and do the data support the conclusions?

Reviewer #4: Yes

3. Has the statistical analysis been performed appropriately and rigorously?

Reviewer #4: Yes

4. Have the authors made all data underlying the findings in their manuscript fully available?

Reviewer #4: Yes

5. Is the manuscript presented in an intelligible fashion and written in standard English?

Reviewer #4: Yes

Reviewer #4: The authors clarified and answered the remaining question by this reviewer concerning the call of Fig.4 and SFig.9 in text.

**Do you want your identity to be public for this peer review?** For information about this choice, including consent withdrawal, please see our Privacy Policy

Reviewer #4: No

---

## [Editor Report · Acceptance letter]

PONE-D-25-04390R2

PLOS ONE

Dear Dr. Nashimoto,

I'm pleased to inform you that your manuscript has been deemed suitable for publication in PLOS ONE. Congratulations! Your manuscript is now being handed over to our production team.

Kind regards,

on behalf of

Dr. Hodaka Fujii

Academic Editor

PLOS ONE